# Mechanical Behavior of Briquettes Made from a Mixture of Sawdust and Rice Husks for Commercialization

Oscar Araque [1,*], Nelson Arzola [2] and Laura Gallego [3]

1    Department of Mechanical Engineering, Universidad de Ibagué, Ibagué 730001, Colombia
2    Research Group in Multidisciplinary Optimal Design, Departamento de Ingeniería Mecánica y Mecatrónica, Universidad Nacional de Colombia, Bogota 111321, Colombia; narzola@unal.edu.co
3    Planaudi Research Group, Universidad Cooperativa de Colombia, Sede Ibagué—Espinal, Ibagué 730001, Colombia; laura.gallego@campusucc.edu.co
*    Correspondence: oscar.araque@unibague.edu.co; Tel.: +57-608-2760010

**Abstract:** The development of this research work seeks to determine the mechanical behavior of the compacted mixture of pine sawdust and rice husk by varying the mass percentages of these biomasses obtained in briquettes, with the purpose of being marketed. The finite element software ANSYS is used to corroborate the results obtained for the samples named AIO, BIO and CIO with rice husk mass percentages of 25, 50 and 75, respectively. In the computational simulations, Young's moduli of between 651 and 813 MPa and a Poisson's ratio of 0.8 were found for all samples. In compression tests, Von Mises stresses of between 87 and 90 MPa and Von Mises strains between 0.09 and 0.12 m/m were found. Free-fall tests showed von Mises stresses below 4.24 MPa. It was determined that increasing the percentage of rice husk mass in the simulation models increased the value of Young's modulus and compressive strength, this is a positive indication in relation to the strength of the formed briquettes, in case they are required to be transported for commercial purposes, however no defined stress and strain behavior was obtained for the impact tests, since the heights of 2 m and 5 m together with the masses established for the specimens do not cause high impact forces.

**Keywords:** biomass; simulation; Young's modulus; Poisson's coefficient; compression; deformations

## 1. Introduction

Biomass is an organic material that comes from plants and animals, and is a renewable source of energy. This material contains stored energy from the sun. Plants absorb energy from the sun in a process called photosynthesis. When biomass is burned, the chemical energy in the biomass is released as heat. Biomass can be burned directly or converted into liquid biofuels or biogas that can be burned as fuel [1,2].

The constant need for sustainable energy is on the rise due to urbanization, population growth and development. The transformation of biomass resources could increase energy supply and promote the energy mix. One way to harness biomass resources is by using densification, which is the application of pressure to form solid fuels that have advantages for domestic and industrial application [3–5].

Some of the research that has been developed in South America about the use of these residues, which are also called biomass because of their belonging to the group of energy products and renewable raw materials that originate from organic matter, includes the investigation carried out by Cabrales et al. [6].

In this work, the mechanical characterization of biomass briquettes made from African palm oil residues was carried out, where the results showed the feasibility of developing products of this type with raw materials available in Colombia [6]. Another piece of academic research developed by engineers Vargas, Atuesta and Sierra focused on reviewing a mixture composed of coal in specific percentages measuring both chemical, energetic and physical properties, focusing on impact resistance. The results showed improvements

by increasing the portion of coal in the mixture in terms of calorific value, but with poor results with respect to the environmental level, with residues well above those indicated by regulations for the production of briquettes or pellets [7]. Several investigations use biomasses or mixtures of these, for example, rice husks, sawdust from different kind of wood such as oak, pine, araucaria, among others, corn residues and some species of long grasses, which contain a high percentage of lignin content which is responsible for improving the physical and mechanical properties during densification.

One of these investigations mechanically evaluated mixtures of lignocellulosic residues after being subjected to densification processes, in order to be physically and mechanically characterized to determine their compressive strength, particle size, moisture and microstructure. Based on these studies, it was concluded that, during the biofuel conformation, an additional binder to the lignin that each of them possessed was not needed, due to the moisture present in them. The bonding bridges were increased when it was activated by the high stresses on the biomass, which guaranteed greater durability and flexibility [8–10].

The researcher Lubwama et al. indicated that in order to evaluate the mechanical behavior of these composite materials, it is necessary to carry out different types of tests [11]. The research article by Lisperguer and Solis [12] studied the effect of pine and raulí wood flour on the mechanical properties of wood-plastic composite materials, when incorporated in percentages of 10% and 20% into the polymeric matrix. These mixtures were made in an extruder-injector machine for single screw plastics. In order to obtain the tensile strength and impact resistance, tests were carried out according to ASTM 256-93 and ASTM 638-00 standards. As a result of these tests, a strong increase in tensile strength was found when the replacement percentage is 10% by weight of wood flour, reaching an increase of 40% more than the pure polymer used (PP). When the weight of the polymer is replaced by 20% of the weight with this biomass, the resistance decreases until reaching values similar to pure thermoplastics. For the impact resistance, it was recommended to use wood filler up to 10% of the weight. Another piece of research, ref. [13], investigated the compressive properties of composites of wood particles and cements. A total of 22 samples of Portland cement (Type I) and yellow pine particles were manufactured. These properties were obtained by performing the compression test in an INSTRON machine with a loading rate of 0.12 cm per minute and following the procedure described in ASTM D1037-78. The deformation-load curves showed significant non-linearity and indicated that this composite has the capacity to absorb energy. In addition, it was determined that the mechanical properties have directional dependencies due to the orientation of the wood particles caused by pressing during manufacture [14].

The study by Gong et al. [15] examines the effect of particle size and wood flour content on the properties of polystyrene, wood-plastic ratios 10:90, 30:70 and 50:50 and particle sizes of 40, 50, 65 and 100 mesh, to evaluate the tensile, flexural and impact bending strength; the tensile test was performed in a universal machine ASMF-100, under ASTM D790-02, with a speed 5 mm/min and a load cell of 900 N. For the bending test, the aforementioned standard was used with a speed of 4 mm/min and a load cell of 900 N. Finally, the impact bending test was carried out under ASTM D 5628-96 using a 0.1724 kg scientific instrument in free fall. The results show that the mechanical properties are strongly influenced by wood flour content and particle size.

The purpose of this research is to study the behavior of the mechanical properties of compacted or densified mixtures when varying the percentage of pine sawdust and rice husk mass, by means of simulation models developed in ANSYS V20, licensed using the finite element method. This will contribute to the selection of suitable mixtures of the above-mentioned biomasses, because the briquettes need high mechanical durability to maintain high quality during transportation, loading and other logistic steps for the different conditions of storage and handling after densification with the purpose of being commercialized [16–18].

Some effective experiences in the use of biomass resources are the projects developed in the province of Hubei China, where densified raw materials have been used for energy production, with the purpose of using the country's agricultural production residues [19].

Other studies have evaluated the technical and economic feasibility of the production of biomass briquettes. In this study, the potential use of waste from mixtures of waste paper, sawdust and carbonized rice husks was determined in informal economy sectors, demonstrating the possibility of economic exploitation for the supply of domestic energy in small businesses [20].

The study in [21] aims to demonstrate the technical and economic feasibility of co-gasification schemes with biomass and pre-combustion of coal and petroleum coke mixtures with biomass available in the Mediterranean area, resulting in a 16% reduction of fossil $CO_2$ emissions with a net power loss of less than 6%, indicating that biomass mixtures become attractive alternatives as an energy alternative.

## 2. Materials and Methods

In order to develop the computational simulations using ANSYS software, it is necessary to characterize the biomasses that compose the specimens that are the object of this study. The mechanical characterization of pine sawdust was extracted from the research carried out by Holm, Henriksen, Hustad and Sonrense [22], in which a differential with mechanical properties identical to those of pine wood is proposed, with the purpose of establishing the mechanical properties of a pine sawdust particle. The results of the physical-mechanical properties are shown in Table 1.

**Table 1.** Mechanical properties of Softwood pine [22].

| Species | Modulus of Elasticity EL (MPa) | Density (kg/m³) | Poisson's Ration |
|---|---|---|---|
| Softwood pine | 6634 | 383 | 0.337 |

For rice husk an average Young's modulus for rice husk close to 2600 MPa was found [23]. For the determination of this magnitude, rice husks were sun-dried in an open field before storage, these were further dried in an oven at 100 °C for 24 h before measurements and stored in a sealed container. The husks were measured using image processing software, finding an average length and width of 7.14 ± 1.68 mm and 1.24 ± 0.27 mm, respectively, from 264 randomly chosen samples, the characteristics of sawdust corresponded to particles with dimensions less than 1 mm [23] and were air-dried. Rice husk properties were measured by performing uniaxial load tests along the rice husks. The tensile test specimen was prepared by bonding both ends of an undamaged rice husk with a high-strength epoxy (Loctite®). The thickness and length of each husk sample was measured using a micrometer. The sample was then loaded into a universal testing machine with a 10 N cell load at a rate of 0.5 mm/s until failure. The stress–strain curves of the rice husks were plotted with the average of all the tests on the rice husks. The results obtained are in agreement with the study conducted by Ziyong Chen, Yangzi Xu, and Satya Shivkumar [24], in which the tensile elastic properties exhibited by seven rice husk varieties were determined.

### 2.1. Determination of Experimental Parameters

The development of the research consists of two work phases: the experimental phase and the simulation phase; the experimental phase consisted of mixing pine sawdust and rice husks, with mass proportions of 25%, 50% and 75% of rice husk biomass, and then compacting it to form briquettes using compression time of 20 s and a heating temperature of 90° Celsius. For each established proportion of the rice husk (combinations) 18 samples were made for the experimental study [25]. For each combination mentioned above, the height, diameter, Young's modulus, Poisson's coefficient and compressive strength were measured, as well as the densities of the biomasses used for the manufacture of specimens.

Table 2 shows the manufacturing parameters, physical and mechanical properties established in the experimentation for the aforementioned samples.

**Table 2.** Physical-Mechanical Properties of briquettes.

| Physical-Mechanical Properties of Briquettes | Samples | | |
|---|---|---|---|
| | AIO | BIO | CIO |
| Mass % of Rice Husks | 25 | 50 | 75 |
| Compression time (s) | 20.00 | 20.00 | 20.00 |
| Compaction temperature (°C) | 90.00 | 90.00 | 90.00 |
| Average Bulk density (kg/m$^3$) | 1124.25 | 1097.08 | 1069.91 |
| Density of pine sawdust (kg/m$^3$) | 396.36 | 396.36 | 396.36 |
| Density of rice husk (kg/m$^3$) | 279.75 | 279.75 | 279.75 |
| Compressive force (KN) | 56.44 | 56.44 | 56.44 |
| Average Young's modulus (MPa) | 719.82 | 818.71 | 917.59 |
| Poisson's Coefficient | 0.13 | 0.13 | 0.13 |
| Average Height (mm) | 47.63 | 48.67 | 49.54 |
| Average Diameter (mm) | 30.18 | 30.17 | 30.20 |

In the simulation phase, the experimental results were used as input data necessary for the development of the simulations in the finite element software. Samples AIO, BIO and CIO containing mass percentages 25%, 50% and 75% of rice husk, respectively, were selected, this is because they are the samples that can be subject to commercialization.

### 2.2. Computational Modelling

For the computational modeling of the mechanical behavior of compacted mixtures of pine sawdust and rice husks in the finite element software ANSYS Workbench V20, the homogenization method was used. This method consists of establishing a Representative Elementary Volume (RVE), which represents the physical, mechanical or thermal behavior of the composite material. The component in charge of performing the homogenization of composite materials is the Material Designer, available in the software.

The computational model used to carry out the homogenization does not evaluate the porosity because it only allows the establishment of two phases (matrix and fiber), therefore, the effects of this will not be reflected in the results to be obtained in this simulation. Consequently, it is expected that the simulated results will have a small difference to the experimental results, but with similar behavior of the mechanical properties. The software defines the size of the RVE, cylindrical fibers are used and the user sets their positions in the three axes [26].

The simulation process considers the following; pine sawdust and rice husk were established as matrix and fiber, respectively, within the composite material to be constructed. This consideration was taken because rice husk particles have a more constant morphology, size and weight than pine sawdust particles. The incidence of the increase in the percentage of rice husk mass on the mechanical properties of the compacted briquettes is determined. The density of pine sawdust and rice husk is entered into the engineering data of the Designer material, and increased according to the compression ratio resulting from the densification process. The mechanical behavior of the matrix was adjusted to the different orientations and positions of the pine sawdust particles, for this reason it was established that its behavior is close to an isotropic material, exhibiting a Young's modulus very similar to the radial Young's modulus of a particle of this biomass.

### 2.3. Simulation Process

In the Engineering Data of the Material Designer add-on, the two materials representing pine sawdust and rice husk were created, with the mechanical properties shown in Tables 1 and 2. In the densification process, the density of the pine sawdust and rice husk mixture increases as the volume change produced by compression occurs, keeping

the mass without significant changes, so it is assumed to be the same before and after this process [27–30]. The following calculations are used to obtain the densities of the biomass after being compressed ($\rho'_{sawdust}$, $\rho'_{rice\ husk}$), the results are shown in Table 3. To achieve this, the density and volume of each specimen $\rho_p\left(\frac{Kg}{m^3}\right)$, $V_p(m^3)$, listed in Table 2, are used, where $A_p$ is specimen area and $\phi_p$ specimen diameter.

$$\rho_p = \frac{m_p}{V_p} \tag{1}$$

$$m_p = \rho_p \times V_p \tag{2}$$

$$m_p = \rho_p \times (h_p \times A_p) \tag{3}$$

When the mass of the specimen $m_p$, is found, the mass of the sawdust ($m_{sawdust}$) in kg, and the mass of the rice husk ($m_{rice\ husk}$) in kg, are calculated, taking into account the mass percentage of each component of the specimen (%m).

$$m_p = m_{sawdust} + m_{husk} \tag{4}$$

$$m_p = m_p \times \%m_{sawdust} + m_p \times \%m_{husk} \tag{5}$$

With the mass of each biomass, the volume of each biomass is obtained ($v_{sawdust}$, $v_{rice\ husk}$) in m³.

$$\rho_{sawdust} = \frac{m_{sawdust}}{V_{sawdust}} \tag{6}$$

$$\rho_{rice\ husk} = \frac{m_{rice\ husk}}{V_{rice\ husk}} \tag{7}$$

$$V_{sawdust} = \frac{m_{sawdust}}{D_{sawdust}} \tag{8}$$

$$V_{rice\ husk} = \frac{m_{rice\ husk}}{D_{rice\ husk}} \tag{9}$$

When these volumes are added together, the volume of the mixture ($V_{mix}$) that will make up the test specimen is determined.

$$V_{mix} = V_{sawdust} + V_{rice\ husk} \tag{10}$$

Finally, the volume change that occurred in the densification process ($\Delta V$) was calculated to find the density of each biomass after being compressed.

$$\Delta V = \frac{V_{mix}}{V_p} \tag{11}$$

$$\rho'_{sawdust} = \rho_{sawdust} \times V \tag{12}$$

$$\rho'_{rice\ husk} = \rho_{rice\ husk} \times \Delta V \tag{13}$$

**Table 3.** Densities of pine sawdust and rice husks upon compaction.

| Samples | $\Delta V$ | $\rho'_{sawdust}\left(\frac{Kg}{m^3}\right)$ | $\rho'_{rice\ husk}\left(\frac{Kg}{m^3}\right)$ |
|---|---|---|---|
| AIO | 3.13 | 1241.41 | 876.18 |
| BIO | 3.34 | 1325.73 | 935.70 |
| CIO | 3.54 | 991.22 | 991.22 |

Type of RVE: By observing the internal organization of the materials that make up the sawdust and rice husk specimens, the RVE Chopped Fiber Composite was selected [31,32].

Geometry of the RVE: For the construction of the RVE it is necessary to determine the volume fraction occupied by the fiber and its geometry. The volume fraction of the fiber is the volumetric portion occupied by the rice husk inside the specimen (see Table 4). It is calculated with Equation (14).

$$\%V_{fiber} = \frac{V_{rice\ husk}}{V_{mix}} \tag{14}$$

**Table 4.** Percentage of fiber volume in the RVE.

| Samples | $V_{sawdust}$ (cm$^3$) | $V_{rice\ husk}$ (cm$^3$) | $\%V_{fiber}$ |
|---------|---------|---------|---------|
| AIO | 71.90 | 33.96 | 0.32 |
| BIO | 47.94 | 67.92 | 0.59 |
| CIO | 23.97 | 101.88 | 0.81 |

The fiber is predefined with cylindrical morphology by the software; therefore, it was guaranteed that the length and diameter for a rice husk defined in the software were similar to the real dimensions. This was achieved by consulting academic articles on the subject. The relationship between the diameter and length of a fiber was found (see Table 5), with the following information: the unit mass of rice husk ($m_{fu}$) is between 2.5 mg and 4.5 mg and average length $L_f$ of 7.14 mm [33,34]. With these data, the diameter of a single fiber ($\phi_f$) mm, was calculated, as well the aspect ratio obtained as the quotient between the length of the fiber and the diameter, as seen below.

$$V_{fu} = \frac{m_{fu}}{\rho_{rice\ husk} \times \Delta V} \tag{15}$$

$$V_{fu} = \frac{\left[\left(\frac{2.5}{10^6}\text{Kg} + \frac{4.5}{10^6}\text{Kg}\right)/2\right]}{279.75\ \frac{\text{Kg}}{\text{m}^3} \times \Delta V} \tag{16}$$

$$V_{fu} = L_f \times \frac{\pi}{4} \times \phi_f{}^2 \tag{17}$$

$$\phi_f{}^2 = \frac{V_{fu}}{L_f \times \frac{\pi}{4}} \tag{18}$$

$$\phi_f = \sqrt{\frac{V_{fu}}{L_f \times \frac{\pi}{4}}} \tag{19}$$

**Table 5.** Diameter and aspect ratio of a fiber in the RVE.

| Samples | $m_{fu}$ (µg) | $\phi_f$ (mm) | $L_f/\phi_f$ |
|---------|---------|---------|---------|
| AIO | 3.5 | 8.44 | 8.46 |
| BIO | 3.5 | 8.17 | 8.74 |
| CIO | 3.5 | 7.94 | 9.00 |

Type of solution: The output data from Material Designer are for orthotropic materials, therefore, three elastic modules ($E_x$, $E_y$, $E_z$) and three Poisson's coefficients ($v_{xy}$, $v_{yx}$, $v_{xz}$) will be obtained for the RVE.

## 3. Results

The compression test is carried out to determine the capacity of the briquettes to support crushing loads. A Shimadzu UH-I 50 kNI universal machine is used for this test, the installation is shown in Figure 1.

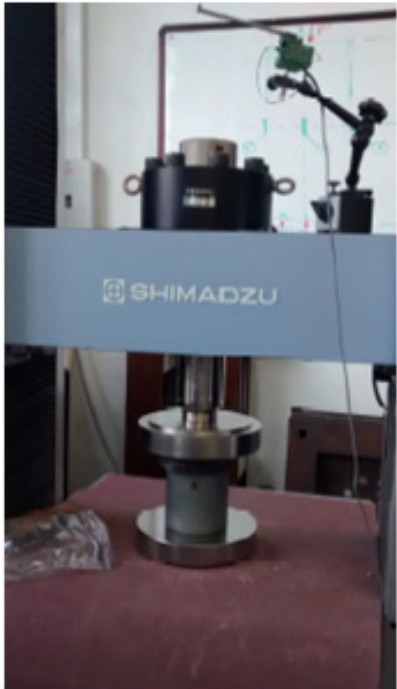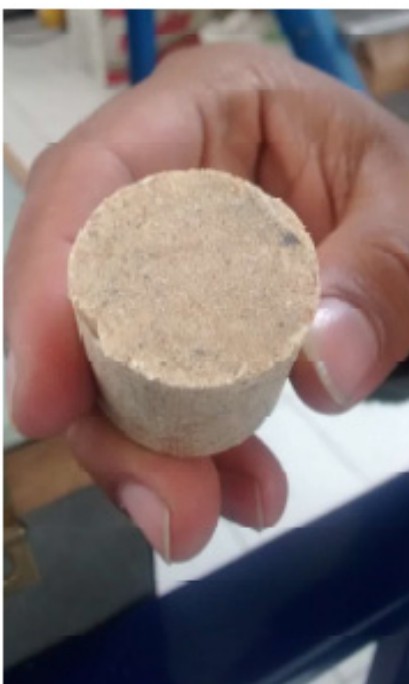

**Figure 1.** Compression test installation.

The following are the results of the compression test for the percentages of rice husk under study, it is observed that the compressive strength is in the range between 9700 and 10,000 kg/mm$^2$, finding a very low dispersion among the data obtained, These measurements are shown in Figure 2.

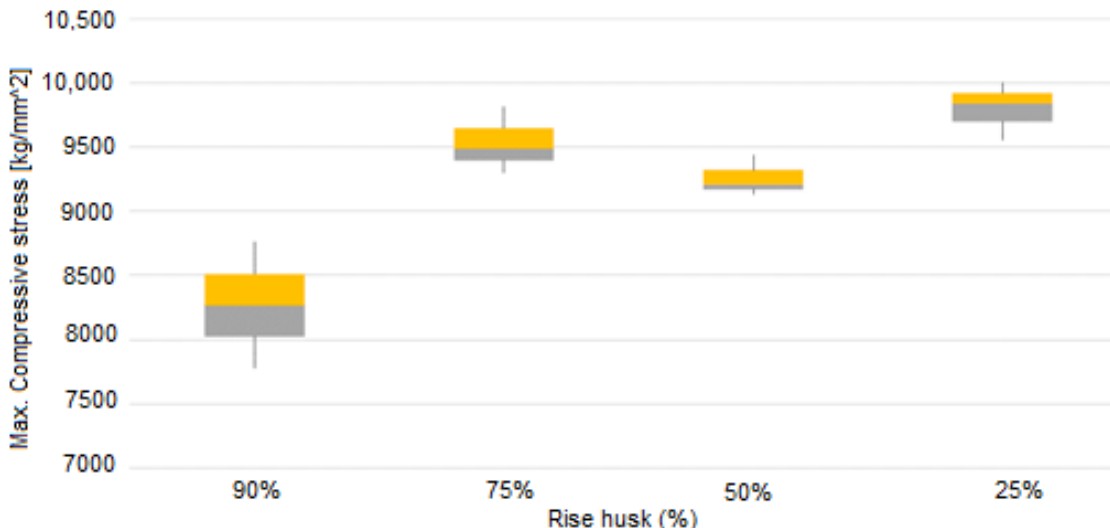

**Figure 2.** Compressive stress results.

Using optical microscopy with an Olympus CX21 equipment, which provides a magnification of $700\times$ ($0.70 \times 1000$), the inclination of the rice husk within the densified material was established, obtaining an approximate value between 30° and 60°, it is identified that the distribution is random and corresponds to the accommodation of the material without external interference, some of the micrographs obtained are shown in Figure 3.

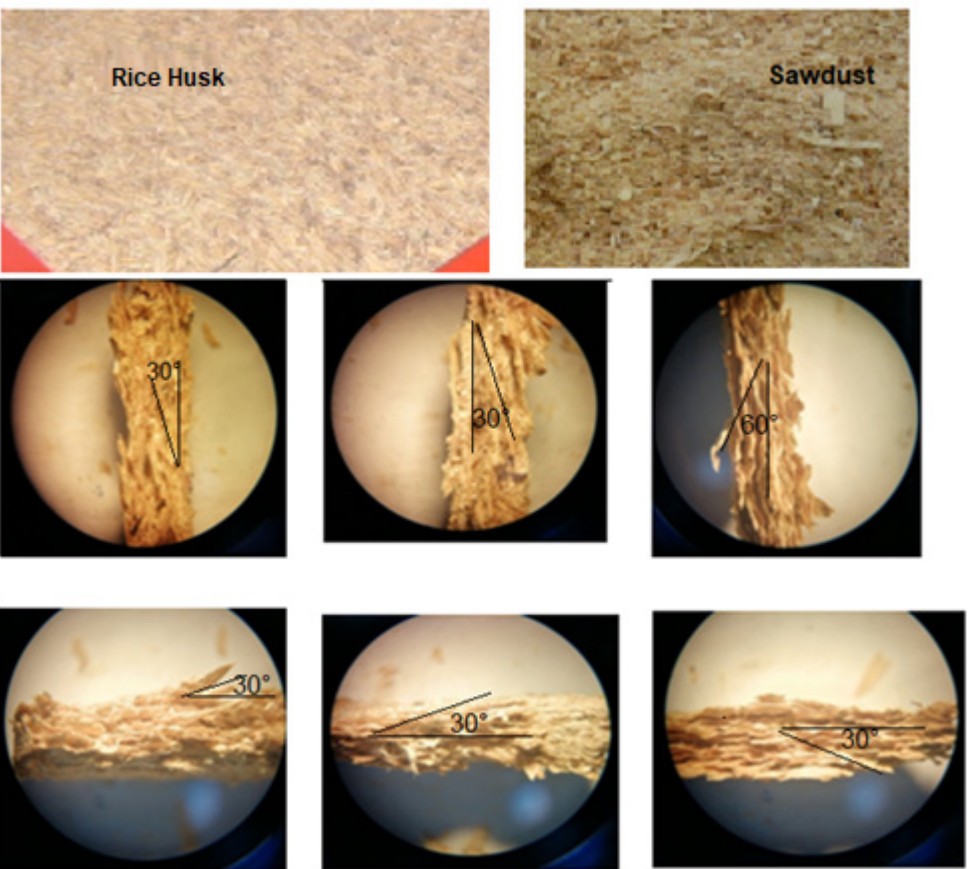

**Figure 3.** Optical micrographs of the densified material in longitudinal direction.

The rice husk inclination angle will be used in the homogenization of the densified material for the construction of the Representative Elementary Volume (RVE) in the Ansys Workbench V20 software.

Using the established parameters, the AIO, BIO and CIO specimens are conformed, guaranteeing that this inclination is present for each of the models, taking into account the importance of its incidence in the values found for Young's modulus and Poisson's coefficient, in the longitudinal axis of the specimen in the experimental phase, of the compacted material of this biofuel.

### 3.1. Simulation of Compression Test

The process of simulating the compression test on the specimens was performed in ANSYS Workbench V20 using Static Structural [35,36] as shown below.

Homogenization of the compacted material: To simulate the mechanical behavior of the compacted sawdust and rice husk mixtures in the ANSYS finite element software, the homogenization method was used. This method consists of establishing a Representative Elementary Volume (RVE), which represents the physical, mechanical or thermal behavior of the composite material. The component in charge of performing the homogenization of composite materials is the Material Designer, available in ANSYS Workbench V20.

Considerations: Rice husk was established as a fiber within the composite material. This consideration was taken into account because rice husk particles have a more constant morphology, size and weight than sawdust particles. Therefore, it is more practical to determine the incidence of the increase in the percentage of rice husk mass on the mechanical properties, if it is evaluated as a fiber in the composite material.

When defining the density of rice husk in the Engineering Data of the Designer material, it was increased according to the compression ratio used in the compaction process. The Chopped Fiber Composite type of RVE was chosen to be used. The orientations

and positions within the RVE of these fibers are random, much like rice husks within the compacted mixture.

Limitations: In the ANSYS add-in used to carry out the homogenization, it does not evaluate the porosity because it only allows the establishment of two phases (matrix and fiber), therefore, the effects of this will not be reflected in the results to be obtained in this simulation.

Simulation Process: The simulation process of the densified biomass mixture was carried out in ANSYS Workbench using the Material Designer, initially the materials were defined; in the Engineering Data cell of the Material Designer add-on, the two materials representing matrix and fiber were created. In addition, the values of the densities of these biomasses after being compressed were recorded. In the compaction process, the density of the biomass mixture increases when the volume change produced by the compression occurs, maintaining the mass without significant changes, so it is assumed that it is the same before and after this process.

Geometry: The specimens were constructed in the software according to the height and diameter for each specimen (AIO, BIO and CIO), obtained during the experimental phase. In addition, two solids were created representing the actuator of the compression testing machine and a fixed contact surface with the bottom face of the specimen, as shown in Figure 4.

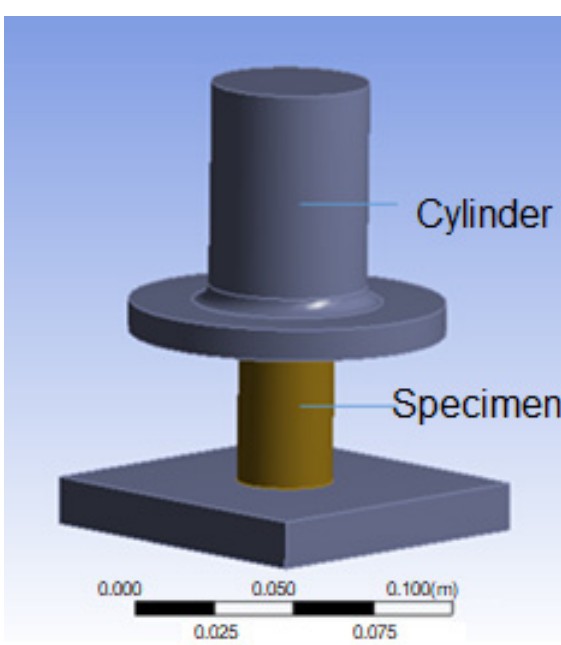

**Figure 4.** General setup of the compression test.

Physical-mechanical properties: Young's modulus, density and Poisson's ratio values are used for each specimen (see Table 2). These properties are transferred to the Static Structural engineering data.

Type of Contact: for the contact between the contact surfaces between the actuator and the specimen, and between the specimen and the fixed surface, rough contact was used because the direction of motion is perpendicular to the contact area between the surfaces, this being important since it allows small slips between these areas.

Boundary conditions: To obtain the stresses and deformation by von Mises criterion exhibited by the pine sawdust and rice husk specimens when compressive loads were applied, a compressive force distributed on the upper face and a fixed restraint at the base of the specimen were established.

The magnitude of the force is dependent on the duration of the test and limited by the compressive strength value of all specimens in the experimental phase, determining a

maximum compressive force $F_{\max}$ with magnitude 56.44 kN. The deformation was found by performing preliminary tests using the $F_{\max}$ with simulation times no longer than a quarter of a minute, finding deformation values very close to each other. With this, the deformation in each simulation was established and the approximate duration of each test proposed in this item was found, using Equation (20).

$$t_{test} = \frac{deformation \left(\frac{m}{m}\right)}{deformation\ velocity \left(\frac{m}{m \times min}\right)} \tag{20}$$

When the duration of the test was obtained, a force as a function of time was created in the Static Structural, with the expressions of Equation (22). The results are shown in Table 6.

$$F_c = \frac{F_{\max test}}{t_{test}} \tag{21}$$

$$F(t) = F_c \times t_{test} \tag{22}$$

**Table 6.** Diameter and aspect ratio of a fiber in the RVE.

| Samples | $deformation\ \left(\frac{m}{m}\right)$ | $t_{test}$ (s) | $F(t)$ (N) |
|---------|----------------------------|----------------|------------------|
| AIO | 0.12 | 1.50 | 37,627 × time |
| BIO | 0.11 | 1.30 | 43,415 × time |
| CIO | 0.10 | 1.16 | 48,655 × time |

*3.2. Free Fall Test*

The objective of this test is to analyze the stresses and deformations presented by pine sawdust and rice husk specimens an instant after impact against a surface, the importance of this study lies in determining the impact strength of the specimens, in order to establish the conditions for their safe storage and transportation. This study consists of letting the specimen fall freely by gravity from a height of 2 m and 5 m, varying the angles of inclination of the lower face of the specimen from 0°, 45° and 90° with respect to the impact surface, in order to analyze the concentration, magnitudes of stresses and deformations (von Mises) in each specimen. The free-fall test specimen simulation process was performed in ANSYS Workbench V20 using Explicit Dynamics as shown below [37,38].

Geometry: The specimens are constructed in the software according to the height and diameter for each sample, taken in the experimental phase (see Table 3). The bottom face of the specimen is also oriented with respect to the plane of the impact surface having the impact angles indicated above, as shown in Figure 5.

Definition of Properties: Young's modulus, density and Poisson's ratio values were taken for the exposed specimens tested. These properties were established in the Engineering data of Explicit Dynamics. Likewise, concrete and structural steel were selected as materials for the impact-bearing surfaces. Their properties are in the materials database of the software.

Loads and restraints: The specimens are only under the effect of the load produced by their mass and gravity (weight) when allowed to fall freely. On the impact surface a fixed clamping is placed in the space so that the effects of gravity do not make it fall, this ensures the impact between these two elements.

Initial velocity: the specimens are 1 m from the impact surface, this height was used for the proposed simulations at heights of 2 m and 5 m. This was possible by imposing an initial velocity to the specimens according to the final velocity of the missing path (1 m and 4 m) for each case of study, in order to reduce the computational load. For this purpose, the Drop Height component in Explicit Dynamics of the ANSYS V20 software was used.

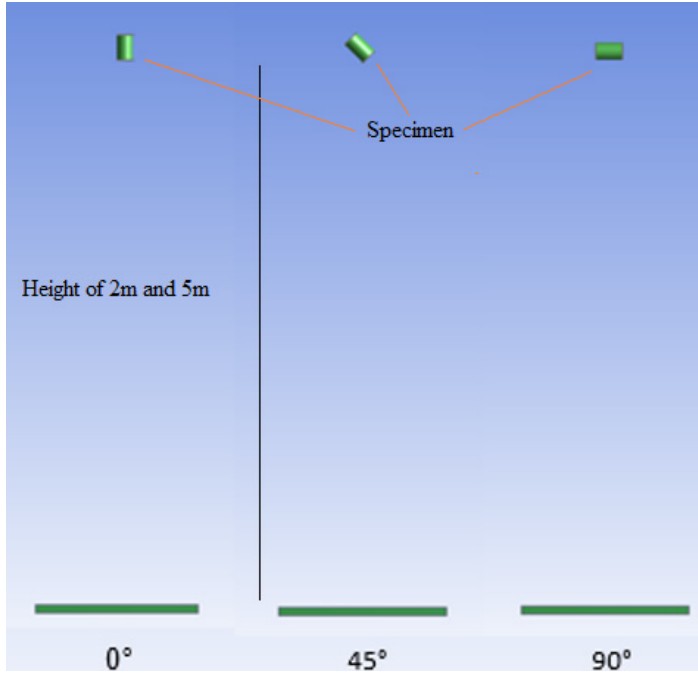

**Figure 5.** General set-up of the free-fall test, using impact angles of 0, 45 and 90°.

## 4. Discussion

### 4.1. Mechanical Behavior of the Densified Material

The geometric configuration for the RVE of the materials that make up the AIO, BIO and CIO specimens were obtained by using the physical-mechanical parameters incorporated as input data for the execution of the homogenization of the biofuel composite material. As can be seen in the RVEs shown in Figure 6a–c, the fiber inclination was established between 30 and 60°, approaching the internal organization of the materials that make up the densified material identified in the experimental phase.

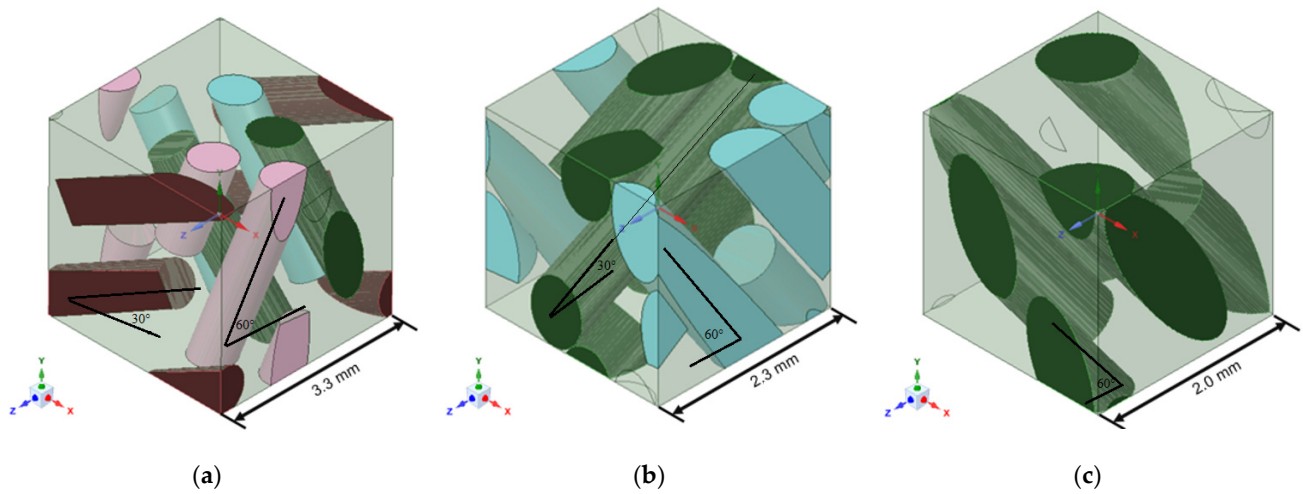

| (a) | (b) | (c) |

**Figure 6.** RVE for the samples, (**a**) AIO; (**b**) BIO; (**c**) CIO.

Modulus of elasticity obtained by experimental simulation are shown in Table 7.

It is observed that the modulus of elasticity found by simulation increases with increasing percentages of rice husk mass (fiber) in the RVE, behaving similarly to the results found experimentally. A difference is observed between the Poisson's coefficients found in the simulation and those obtained experimentally; this is because in the computational model it was not possible to incorporate the porosity, because in the Designer material

complement only two phases are established in the composite: matrix and fiber. These results however are related to the behavior observed in the research [39].

**Table 7.** Mechanical properties of RVE.

| Samples | Young's Module $E_y$ (MPa) | Poisson's Coefficient $v_{xy}$ |
|---------|----------------------------|--------------------------------|
| AIO | 651.61 | 0.08 |
| BIO | 758.86 | 0.08 |
| CIO | 813.10 | 0.08 |

### 4.2. Compressive Strength

The values of stresses and von Mises deformations obtained by computational simulation for the AIO, BIO and CIO samples are shown in Table 8.

**Table 8.** Von Mises stresses and Equivalent Strains resulting from the compression test.

| Samples | Equivalent Stress Von Mises Max. (MPa) | Equivalent Strain Von Mises Max. (m/m) |
|---------|----------------------------------------|----------------------------------------|
| AIO | 90.09 | 0.12518 |
| BIO | 88.41 | 0.10802 |
| CIO | 87.82 | 0.09573 |

Regarding the magnitude of the compressive stresses for the AIO, BIO, CIO samples, it can be seen that there is a difference of approximately 8% with respect to the values obtained from the simulated model, this difference is mainly due to the fact that it was not possible to include the porosity in the simulation models, and when the briquettes are subjected to compression loads, these voids behave as stress concentrators and initiators of cracks that cause the specimens to fracture during the compression test.

The von Mises stress distribution that the specimens presented in the compression test in the software have a behavior similar to that shown in Figure 7a. It is observed that the highest stresses are on the edge of the upper and lower face of the specimens, these are generated by the contact with the fixed base and the solid that compresses this biofuel [40,41]. Figure 7b shows the image of a specimen experimentally failed during the compression test.

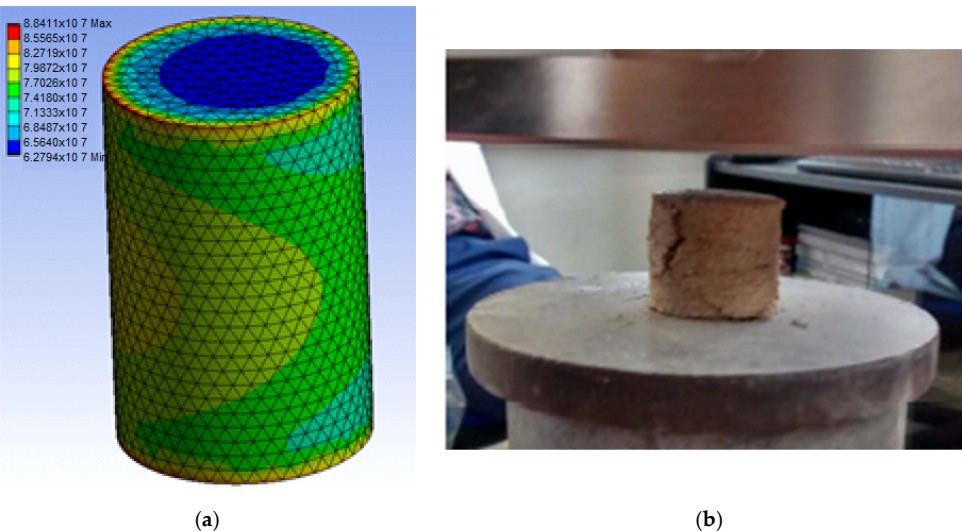

(**a**) (**b**)

**Figure 7.** (**a**) Von Mises stress distribution for the simulated sample; (**b**) Briquette failed in experimental test.

*4.3. Impact Tests*

The results obtained from the free-fall impact tests are shown in Table 9, which shows the equivalent stresses and von Mises' deformations for each configuration proposed in the computational model.

**Table 9.** Von Mises' stresses and strains obtained in the impact test.

| Samples | Height of Fall (m) | Impact Angle | Von Mises Stress Max. (MPa) | | Von Mises Strain Max. (m/m) | |
|---|---|---|---|---|---|---|
| | | | Concrete | S. Steel | Concrete | S. Steel |
| AIO | 2 | 0° | 1.412 | 1.527 | 0.001962 | 0.002122 |
| | | 45° | 0.146 | 0.069 | 0.000301 | 0.002675 |
| | | 90° | 0.886 | 0.376 | 0.001234 | 0.000523 |
| | 5 | 0° | 4.182 | 3.845 | 0.005811 | 0.005342 |
| | | 45° | 0.306 | 0.276 | 0.001119 | 0.000666 |
| | | 90° | 2.160 | 1.306 | 0.003004 | 0.001823 |
| BIO | 2 | 0° | 1.360 | 1.359 | 0.001662 | 0.001660 |
| | | 45° | 0.141 | 0.141 | 0.000229 | 0.000230 |
| | | 90° | 0.812 | 0.813 | 0.000992 | 0.000993 |
| | 5 | 0° | 4.234 | 4.236 | 0.005172 | 0.005174 |
| | | 45° | 0.180 | 0.180 | 0.000732 | 0.000733 |
| | | 90° | 2.1780 | 2.182 | 0.002661 | 0.002666 |
| CIO | 2 | 0° | 2.3013 | 2.300 | 0.002508 | 0.002507 |
| | | 45° | 0.1558 | 0.156 | 0.000210 | 0.000210 |
| | | 90° | 0.8250 | 0.825 | 0.000900 | 0.000901 |
| | 5 | 0° | 3.1178 | 3.122 | 0.003398 | 0.00340 |
| | | 45° | 0.2228 | 0.223 | 0.000499 | 0.000499 |
| | | 90° | 1.7178 | 1.7220 | 0.001873 | 0.001877 |

For the magnitudes of the von Mises stresses and strains found from the free fall tests for the two proposed heights, the following was analyzed: the mechanical energy ($E_m$) is the sum of the kinetic energy ($E_k$) and the potential energy ($E_p$) as shown in the following expression.

$$E_m = E_C + E_P \tag{23}$$

It is known that the potential energy just before the impact is zero; therefore, the potential energy just before the impact is zero.

$$E_m = E_C \tag{24}$$

Knowing that the kinetic energy is the work needed to accelerate a body or, in mathematical terms, is the product between the mass of the specimen and the integral of the acceleration, we obtain:

$$E_m = \frac{m_b * v^2}{2} \tag{25}$$

The final velocities ($v$) that the specimens reach before touching the impact surface are governed by the equation established for the falling motion, starting from their maximum height value with zero initial velocity, as shown below. The final velocities ($v$) that the specimens reach before touching the impact surface are governed by the equation established for the falling motion, starting from their maximum height value with zero initial velocity [42,43], as shown below:

$$E_m = \frac{m_p * \left( \sqrt{2gh} \right)^2}{2} \tag{26}$$

$$E_m = \frac{m_p * g * h}{2} \tag{27}$$

where $m_p$ is the mass of the specimen, $g$ is gravity and $h$ is the height between the specimen and the impact surface. Equation (27) shows that as the free fall height increases, the mechanical energy during the impact will be greater, and with it the impact force, within the proposed study model; as a consequence, the von Mises stresses and deformations will be greater for the 5 m free fall test with respect to the results obtained in the 2 m tests, this behavior is shown for the two impact surfaces, and in each of the specimens, as shown in Figure 8.

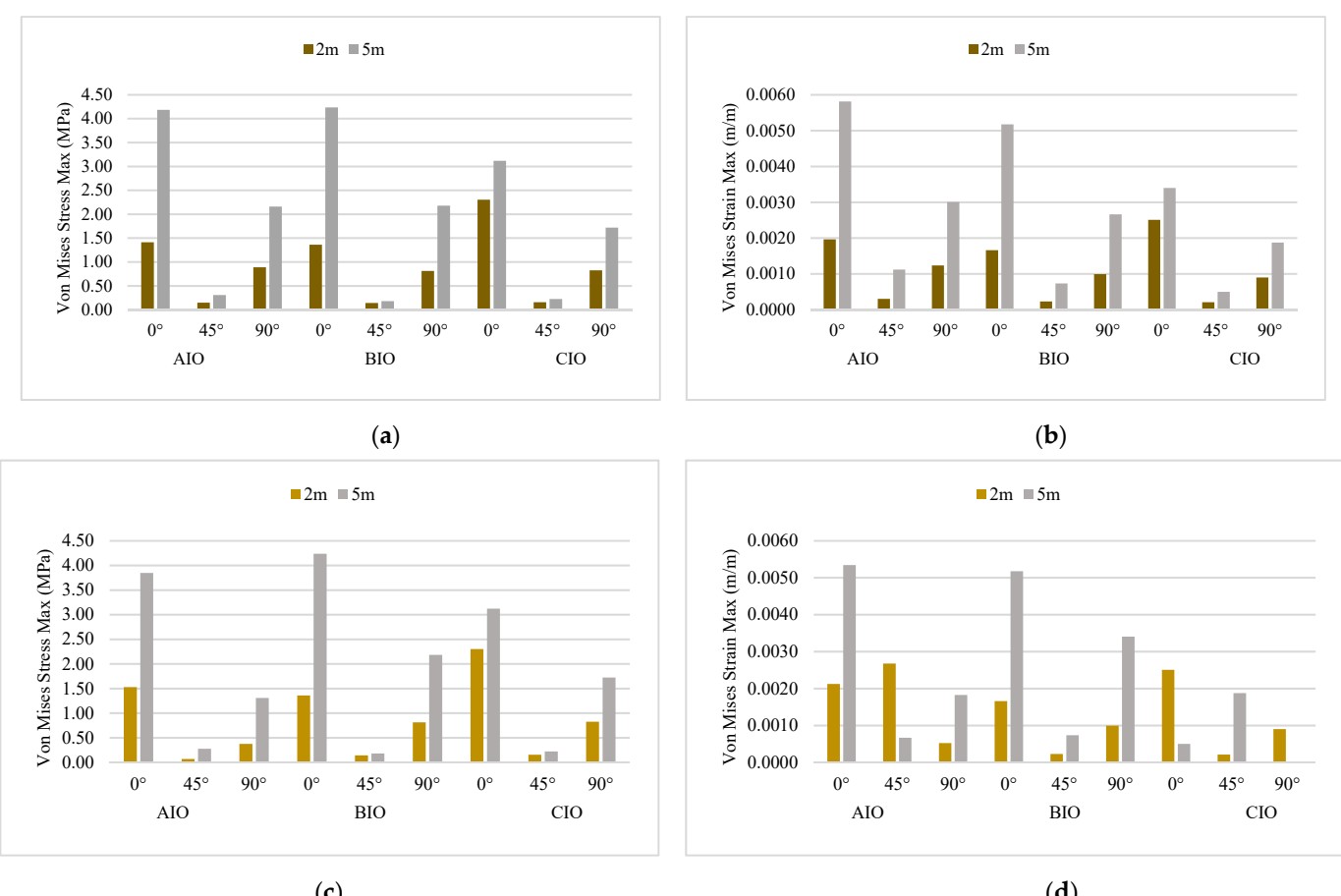

(a)

(b)

(c)

(d)

**Figure 8.** Maximum stresses and deformations in the impact test. (**a**) Maximum stresses in impact test with concrete impact surface; (**b**) Maximum deformations in the impact test with concrete impact surface; (**c**) Maximum stresses in the impact test with a structural steel impact surface; (**d**) Maximum deformations in the impact test with structural steel impact surface.

The results of the impact tests presented by the AIO, BIO and CIO specimens showed that positioning the bottom face of the specimen at 0° with respect to the impact surface, would produce higher von Mises stresses and deformations and the simulations using an impact angle of 45° obtained the lowest values; this behavior is similar in the results obtained for the two impact surfaces. For example, for the impact simulations between the AIO specimen and the concrete impact surface at 2 m height, maximum von Mises stresses of 1.41, 0.15, 0.89 MPa were obtained with impact angles of 0°, 45° and 90°, respectively. Similar behaviors were observed for the structural steel surface.

The stresses presented by the AIO, BIO and CIO specimens did not exceed the elastic limit or failure stress of the specimen, which for the computational modeling was established close to and lower than the value found experimentally for the compressive strength,

therefore, when evaluated by the von Mises criterion this biofuel would not fail with the conditions proposed in each simulation model.

The developed research did not address the properties related to burning briquettes, however, in the specialized literature, researches have been found that use rice husk concentrations similar to those used in the developed study, in the research [44], gross calorific value (MJ/kg) has been determined using the standardized norm DIN 51900, in this it was found that for concentrations of rice husk of 100% the gross calorific value was 17.04 (MJ/kg), for concentrations of rice husk 75% and rice straw 25%, the gross calorific value was 17.37 (MJ/kg), and for concentrations of rice husk 30%, rice straw 60% and rice husk ash 10%, the gross calorific value was 16.92 (MJ/kg), finding minor differences in the energetic power for the different concentrations.

In the work developed by Jamradloedluk and Wiriyaumpaiwong [45], the use of rice husk for the production of briquettes in combination with bagasse has been proposed, obtaining an average combustion temperature of 115 °C for briquettes with 80% rice husk and 100 °C for briquettes with 60% rice husk and 85.5 °C for briquettes with 20% rice husk. Additionally, it was indicated that the mechanical properties, density, ultimate stress and toughness increased with an increased mixing ratio (rice husk/charcoal quantity).

The development of the research allowed the establishment that the computational simulations contrast with the results found experimentally, in relation to the magnitudes of the limit work effort, before breakage, and in this way the specimens formed according to the estimated mixtures of 25%, 50% and 75% of rice husk mass, can guide the scientific community and marketers of biomass briquettes, in relation to the most appropriate percentage combination for the formation of briquettes from the mixture of rice husk and pine sawdust.

This research incorporates the experimentation of mixtures of rice husks and pine sawdust, established from the previous bibliographic review on the subject studied, computational analysis models have been developed using ANSYS V20 software, which have been validated from the experimental results, The main conclusions are the behavior of efforts and deformations of the analyzed mixtures with the purpose of establishing which of these has better mechanical properties that favor their conformation and transport in the event of being commercialized. The methodological application of this study can serve as a basis for other biomass mixtures that can be used in the regions of origin.

## 5. Conclusions

When comparing the experimental and simulated results of Young's modulus, relative errors were found between 7.31% and 11.39%. The behavior obtained in the experimental and simulated tests of the AIO, BIO and CIO samples corroborates that increasing the percentage of rice husk mass increases the Young's modulus, obtaining samples with better mechanical characteristics for their commercialization, storage, transport and handling, within their useful life as biofuel.

It is observed that the specimen with a higher rice husk content shows a higher modulus of elasticity. It was also evidenced that the longitudinal modulus of elasticity increased in relation to the rice husk content in the biomass mixture. This relationship between the increase in husk mass and the longitudinal Young's modulus was produced by the orientation range of the fibers (30 to 60°) within the matrix.

The results obtained in the compression tests showed von Mises stresses between 87.89 and 90.09 MPa, obtaining values very close to each other, and being very similar to the behavior obtained in the compressive strength tests that were performed experimentally. However, the higher percentage of husk mass, the smaller the equivalent von Mises deformations will be.

In the simulation model, the contact between the upper and lower surfaces of the specimen influenced the origin of the highest von Mises stresses in this region, because when the compressive load is applied on the longitudinal axis, the specimen increases its cross-sectional area, causing slippage between the specimen surface and the actuator surface.

The results of the free fall test presented by the AIO, BIO and CIO specimens showed that positioning the lower face of the specimen at 0° with respect to the impact surface would produce higher von Mises stresses and deformations than those presented in the simulations performed for specimens that used an impact angle of 45°, obtaining the lowest values; this behavior was similar for the concrete and steel surfaces.

The briquettes that presented the best mechanical behavior for commercialization were those with a 75% rice husk content, obtained with a compression force of 56.4 kg during a time of 20 s, at a temperature of 90 °C; these results were corroborated by experimental tests and computer simulations.

For the development of the computational simulation, the material designer add-on of the ANSYS V20 software was used, for which two interfaces representing pine sawdust and rice husk were selected. This is a novel procedure, since it allows the determination of properties separately for each material, facilitating the formation of a Representative Elementary Volume (RVE), which represents the physical, mechanical or thermal behavior of the composite material.

In general, there are not many sources of specialized information related to the behavior of briquettes obtained from the mixture of rice husks and pine sawdust. The experimental basis of the study, supported by computer simulations, makes the results obtained become a primary source of information in relation to the lack of knowledge on the subject under study. This enhances the commercial decisions to be taken in this respect.

**Author Contributions:** O.A. conceived, designed and performed the experiments, analyzed the data and wrote the paper; N.A. contributed in the analysis tools and provided technical advice, L.G. contributed with methodology and resources. All authors have read and agreed to the published version of the manuscript.

**Funding:** This research received no external funding.

**Institutional Review Board Statement:** Not applicable.

**Informed Consent Statement:** Not applicable.

**Acknowledgments:** The authors would like to thank the Research Department at the Universidad de Ibagué, Universidad Nacional de Colombia and the Universidad Cooperativa de Colombia for their support provided for the development of this research.

**Conflicts of Interest:** The authors declare no conflict of interest.

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
