# Peer review of "Mechanical Behavior of Briquettes Made from a Mixture of Sawdust and Rice Husks for Commercialization"

_resources, doi:10.3390/resources11030032_

Round 1

Reviewer 1 Report

The introduction is an interesting background for the further described empirical research.
The general description of the research methodology is clear and does not raise doubts.
The test results are presented well and their interpretation is correct. It is also noticable that they refer to the latest scientific literature.

It is worth to underline that the research subject, obtained results and the conclusions are of great utilitarian importance.

Author Response

The authors are grateful for the reviewer's favourable opinion, the manuscript demonstrates the development and results obtained during the research.

Reviewer 2 Report

In this work, the authors studied the mechanical behavior of the compacted mixture of pine sawdust and rice husk by varying the mass percentages of these biomasses using ANSYS finite element software. 
It isn’t very original, but the topic is very interesting, and it is a crucial way to valorize residual biomass. 
Before the publication, some aspects need minor improvements such as: 
1)    Introduction: how this work can help to improve the performances of the residual biomass by gasification, pyrolysis, combustion, is not clear.
2)    Introduction: the authors should describe the advantages of knowledge of the mechanical behavior of solid biomass for its valorization (citing the work Sofia, D., Giuliano, A., Barletta, D., 2013. Techno-economic assessment of co-gasification of coal-petcoke and biomass in IGCC power plants. Chem. Eng. Trans. 32, 1231–1236. https://doi.org/10.3303/CET1332206);
3)    Results: Figure 2 is difficult to read, maybe increase the size of the characters and improve the scale of axes;
4)    Results: scientific discussion on the results is not present in the paper. The research application of this study isn’t clear. Authors have to deepen the potential applications of their research, also extending the same analysis to other systems;
5)    Conclusion: in conclusion, authors have to write more about the lack of the models in terms of the prevision of the mechanical behavior of the briquettes.
6)    In general, the authors should describe the advantages of the adopted approach and obtained results also considering the potential extension of the work to other systems/biomass/shapes.

Author Response

The authors are grateful for the recommendations made by the reviewer, which have been incorporated into the attached manuscript with change control and indicated in the response to reviewer 2.

Reviewer 3 Report

Major Revision:

This paper aims at investigating the properties and the mechanical behavior of the briquettes made from mixture of pine sawdust and rice husk. The authors carried out the experiments by applying two methods, an experimental and through simulations.

The research paper can be improved in order to meet the aim and the scope, as well as, the high academic standards of the ‘Resources’ Journal. Hence, the following specific improvements should be made, before accepting the paper for possible publication to the Journal.

More specifically:

  • The abstract should be improved and be more clear about the results and the type of commercial product (from rice husk or pine sawdust and why)
  • Since the porosity could not be taken into account, you have to explain why this issue will not affect the results
  • In lines 41, 64,83, 527 please write the authors of the references
  • In lines 118-121, except the pretreatment and the drying process of rice husk, you have to refer the sawdust pretreatment accordingly (if not referred)
  • It would be a good idea to add the preferable particle size of the two materials and the way this affects the homogenization of the final sample for briquetting
  • In the conclusion section, it is recommended to add few lines about the method of the simulation used as this is the “innovation” of the study. The presentation of the results by selecting the appropriate mass proportion it is not something innovative to support a research study. However, if the purpose is to highlight that the simulation software helps the ideal choice of the proportions, that should be referred in a clearer way
  • Since the scope of the study is for commercialization, it would be helpful to identify and propose the minimum proportion of the two materials to be used in a briquetting plant for being sustainable, and if there are these quantities available in the region

Author Response

The authors are grateful for the recommendations made by the reviewer, which have been incorporated into the attached manuscript with change control and indicated in the response to reviewer 3.

Round 2

Reviewer 3 Report

Thank you for the application of the recommended changes to your manuscript.